# Genetic Diversity in *Coffea canephora* Genotypes via Digital Phenotyping

**DOI:** 10.3390/plants14182814

**Published:** 2025-09-09

**Authors:** Priscila Sousa, Henrique Vieira, Eileen Santos, Alexandre Viana, Fábio Partelli

**Affiliations:** 1Department of Plant Science, Universidade Estadual Norte Fluminense, Campos dos Goytacazes, Rio de Janeiro 28013-602, Brazil; henrique@uenf.br (H.V.); pirapora@uenf.br (A.V.); 2Research Center, Universidade Estadual do Mato Grosso, Estudo e Desenvolvimento Agroambiental, Tangará da Serra 78300-000, Brazil; eileen.azevedo@unemat.br; 3Department of Plant Science, Universidade Federal do Espírito Santo, São Mateus 29932-540, Brazil

**Keywords:** phenotyping, conilon, GroundEye^®^, genetic divergence, bean geometry, high-throughput phenotyping, coffee breeding, heritability, Ward-MLM

## Abstract

*C. canephora* exhibits high genetic variability, and to estimate this variability, morphological descriptors associated with coffee quality are used. Bean size is a physical trait of great importance for coffee classification. Manual classification is known to be inaccurate and time-consuming, which is why researchers have adopted digital imaging techniques to improve classification efficiency. The objective of this study was to quantify the genetic diversity in 43 *C. canephora* clones using the Ward-MLM strategy and to estimate genetic parameters and correlations from digital phenotyping of beans and cherries. The experiment was conducted on a crop consisting of 43 *C. canephora* genotypes, where the cherries were manually pulped and dried until they reached 12% moisture content. Using GroundEye^®^ equipment, four replicates of 50 beans and cherries were evaluated for each treatment, and the software generated spreadsheets with the results of the geometric traits. To determine the existence of genetic variability among the genotypes, the data obtained were subjected to analysis of variance, estimation of genetic parameters, Ward-MLM analysis, and Pearson correlation. The genotypic variance was higher than the environmental variance for all variables analyzed, both for beans and cherries, indicating that the genotypes evaluated have high genetic variability. The greatest genetic distance was observed between groups I and IV, suggesting favorable conditions for crosses between the genotypes of these groups. Phenotypic correlation analysis revealed significant positive and negative correlations between the variables. Digital seed analysis successfully detected genetic divergence among the 43 *C. canephora* clones. The variables ‘area’, ‘maximum diameter’, and ‘minimum diameter’ are the most suitable for selecting genotypes with larger beans.

## 1. Introduction

*Coffea canephora*, popularly known as Conilon and Robusta, is the second most produced coffee species in the world. Brazil is the second-largest producer of this species, with an estimated 17.1 million bags projected for 2025 [1]. Thousands of people consume coffee daily; the drink carries various nuances and characteristics, and the coffee market is constantly seeking new ways to offer it, whether through blends, espressos, or gourmet coffees. Conilon coffee is widely used in blends with Arabica coffee and in the instant coffee industry due to its higher soluble solids content [2].

*Coffea canephora* is a species that reproduces by obligate cross-pollination due to its gametophytic self-incompatibility [3]. As a result, it exhibits high genetic variability, leading to plants with different potentials [4,5].

Understanding the genetic divergence of a population is essential in any breeding program, as it aids in differentiating accessions, identifying contrasting genotypes for future crosses, and establishing a database to identify potential groups with a higher degree of heterosis [6].

To estimate genetic variability, morphological descriptors of various plant structures can be used, including those related to the leaf, flower, cherry, and seed [7]. Among the morphological descriptors associated with coffee quality, bean size is a key physical trait for coffee classification during processing and the formation of homogeneous lots, as larger beans are generally associated with higher-quality and more valuable coffees.

Since manual classification is known to be inaccurate and time-consuming, researchers have increasingly adopted digital imaging techniques to enhance classification efficiency. Imaging techniques are considered the most effective method for bean classification, and with advances in computer technology, it has become possible to evaluate morphological traits of seeds and beans—such as color, texture, and geometry—through digital phenotyping. This approach is a fast, reliable, and non-destructive technique for quantifying genetic divergence [7,8].

Digital phenotyping offers significant advantages over traditional methods by enabling objective, standardized, and large-scale analysis of morphological traits that would be difficult, time-consuming, or prone to errors when assessed manually. This technology allows for the collection of precise and reproducible data, accelerating the screening and selection of genotypes with superior potential in breeding programs. Furthermore, digital phenotyping reduces evaluation costs and time, contributing to the integration of phenotypic information with genomic data in modern plant breeding strategies.

Venora et al. [9], also reported that image analysis requires less than a minute for scanning and measurement, making it a highly reproducible technique.

In 2011, the company Tbit developed the Groundeye^®^ System Mini to capture images and generate histograms and graphs that facilitate the analysis of seeds, beans, and seedlings [10,11]. Cherry and bean morphometry provide valuable information for phenotyping and genetic characterization of plant species with economic potential.

Abreu et al. [12] conducted a computerized analysis to evaluate the physiological quality of coffee seeds subjected to drying and found that image analysis using the SAS program (Groundeye^®^) is a viable and promising alternative for assessing the viability and vigor of coffee seeds and seedlings. The application of image processing in agriculture enhances efficiency and precision in practices such as classification and inspection, reducing data uncertainty [13].

In breeding programs, certain tools facilitate the breeder’s work, such as genetic parameter estimation and correlation analysis, as these estimates serve as a selection strategy and are representative of the population being studied. Therefore, the objective of this study was to quantify the genetic diversity in 43 *C. canephora* clones using the Ward-MLM strategy and to estimate genetic parameters and correlations through the digital phenotyping of beans and cherries.

## 2. Results

### 2.1. Estimation of Genetic Parameters in C. canephora Cherries and Beans

Nine traits related to the geometry of *C. canephora* beans and cherries were selected for divergence analysis to identify genotypes with larger cherries and beans.

There were significant differences among genotypes for all bean-related (Table 1) and cherry-related (Table 2) variables analyzed. Thus, it can be inferred that there is potential for selecting genotypes with larger bean sizes based on the geometric variables analyzed.

Consequently, the values for broad-sense heritability were high—above 80% for almost all traits, except for sphericity in cherries. For the variable ‘rectangularity’ in both beans and cherries, nearly all the observed variation was of genetic origin, resulting in heritability values close to 100%. This suggests that these traits are less influenced by environmental factors and may yield effective results when selection is based solely on phenotype.

The Variation Indexes exceeded unity for all traits, except for sphericity in cherries (0.783), indicating a greater environmental influence on this variable compared to the others. This can be attributed to the fact that the size and shape of cherries—and consequently the beans—vary according to environmental conditions during their development. On the other hand, the highest Variation Index values were found for bean area and rectangularity, as well as for cherry rectangularity and maximum diameter, indicating that selection for these traits offers the most favorable conditions for immediate genetic gains.

### 2.2. Divergence Between C. canephora Genotypes Based on Digital Phenotyping

According to the pseudo-F and pseudo-T^2^ criteria, in combination with the likelihood profile, the ideal number of groups was determined to be four. This grouping corresponded to the point where the greatest increase in the logarithmic function occurred, with the highest absolute value observed in the fourth group (Figure 1).

The formation of four groups demonstrates that there is genetic variability to be explored within the population and that the use of geometric variables measured through digital seed analysis was effective in quantifying genetic divergence.

Group I consisted of 17 genotypes; group II, 13 genotypes; group III, 9 genotypes; and group IV, 4 genotypes. The traits that contributed most to genetic divergence based on the first canonical variable were bean sphericity (56%), irregularity (57%), and sharpness (50%) (Table 3).

Group I was characterized by beans with the highest rectangularity, the greatest number of corners, and the highest sharpness, as well as cherries with greater sphericity (Table 4). Most genotypes in group I showed medium to high productivity and yield, with genotypes AD1 and L80 standing out, each exceeding 100 bags/ha (Table 4).

Group III was distinguished by genotypes with the largest maximum diameter of beans and cherries, along with the largest bean perimeter. In terms of productivity and yield, genotypes A1 and Peneirão stood out, producing 108.19 and 99.22 bags/ha, respectively.

Group IV comprised genotypes with higher circularity of beans and cherries, greater bean sphericity and irregularity, and greater rectangularity of cherries. Within this group, the Clementino genotype recorded the highest productivity, with 97.86 bags/ha.

These results are fundamentally important for selecting the most promising genotypes in terms of cherry and bean size, aiming to perform crosses focused on obtaining *C. canephora* genotypes with larger beans combined with high productivity.

The first two canonical variables accounted for 82.37% of the total variance, allowing for a satisfactory explanation of the variability among *C. canephora* genotypes in a two-dimensional scatter plot (Figure 2).

The greatest genetic distance was observed between groups I and IV, indicating an advantageous condition for crosses between genotypes from these groups (Table 5 and Figure 2). The distance between group IV and the other groups further supports the potential for obtaining productive genotypes with larger beans through crosses among divergent groups to exploit heterosis.

The purpose of this study was to select genotypes with larger beans and advantageous agronomic traits, aiming to add value to the commercialization of beans and cherries. Therefore, the selection of superior genotypes for these characteristics, particularly bean and cherry size, may contribute to the development of a new cultivar.

### 2.3. Correlation Between Beans and Cherries

Phenotypic correlation analysis revealed significant positive and negative correlations among the variables (Figure 3). In this study, significant correlations were observed within the moderate and strong classifications.

Among the 45 pairs of combinations for the ten geometric variables evaluated, 21 showed significant genotypic correlation at the 1% probability level, of which 17 were positive. Positive genotypic correlations indicate that selection aimed at improving one trait will also enhance the other. This information can facilitate the selection process for traits that are difficult to measure or identify and that may also exhibit low heritability.

Moderate positive correlations represented 6.66% of the total and included: maximum bean diameter × minimum bean diameter (0.57), minimum bean diameter × maximum cherry diameter (0.50), and maximum bean diameter × minimum cherry diameter (0.53). These values clearly show that larger maximum and minimum diameters in cherries are associated with larger bean diameters.

Moderate negative correlations were found between bean sphericity × cherry perimeter (−0.44) and bean irregularity × cherry perimeter (−0.44). Strong negative correlations were observed between maximum bean diameter × cherry circularity (−0.57) and cherry circularity × maximum cherry diameter (−0.60).

Strong positive correlations accounted for 8.88% of the total. These indicate that increases in area, maximum and minimum diameter, sphericity, and irregularity positively influence the size of *C. canephora* beans and cherries. This supports the selection of genotypes with larger beans, as larger cherries tend to produce larger beans. This relationship is especially evident in the correlation between maximum cherry diameter and maximum bean diameter (0.92), which is considered a very strong correlation.

The GroundEye^®^ system enabled the processing of detailed data on the size of *C. canephora* cherries and beans—information that could not be accurately obtained with the naked eye or even with digital calipers. The data extracted from the software are highly precise, secure, and reliable, and not subject to researcher bias. Furthermore, the information is stored in a database, allowing for easy access and further analysis.

It is recommended that selection focuses on increasing area, maximum diameter, and minimum diameter. In addition to their direct positive effect on productivity, these traits indirectly enhance the quality of *C. canephora* beans, as larger beans are visually more appealing and considered higher quality by consumers.

## 3. Discussion

These results demonstrate broad genetic variability among *C. canephora* genotypes, an essential factor for the success of breeding programs. The predominance of genotypic variance over environmental variance reinforces the feasibility of selecting superior genotypes based on the geometric traits of cherries and beans.

The high heritability values observed, particularly for rectangularity and maximum diameter, indicate that these traits are predominantly genetically controlled and less affected by environmental conditions. Therefore, phenotypic selection can be conducted efficiently, leading to consistent genetic gains.

The Variation Index, which exceeded unity for most variables, highlights traits such as bean area and rectangularity, and cherry rectangularity and maximum diameter, as promising targets for selection, as also noted by Nascimento et al. [15].

Ward-MLM analysis identified four distinct groups, confirming the presence of significant genetic variability within the population. The genetic distance between groups I and IV was particularly noteworthy, suggesting the potential for crossbreeding between individuals from these groups to exploit heterosis and maximize genetic gains.

Crosses between Group I (high-yielding, angular beans) and Group IV (high circularity and sphericity) genotypes may generate transgressive segregants combining favorable productivity and bean geometry traits.

The composition of the groups revealed genotypes of agronomic and commercial interest. In group I, genotypes AD1 and L80 exhibited high productivity (>100 bags/ha), along with beans showing greater rectangularity and cherries with greater sphericity. These results are consistent with the findings of Partelli et al. [16], who, when evaluating the same population, reported high productivity in genotypes such as Z36, Ouro Negro 2, P2, and AD1, which also stood out for cherry sphericity.

Furthermore, studies by Silva et al. [17] on root trait diversity in these same genotypes indicated that genotype P2 has greater root volume, particularly in the superficial soil layers, which may contribute to its ability to produce cherries and beans with larger dimensions.

The correlations observed among geometric traits reinforce the feasibility of indirect selection, especially between cherry and bean characteristics. The strong positive correlation between the maximum diameter of cherries and beans (0.92) indicates that larger cherries tend to produce larger beans, allowing for greater efficiency in the selection process.

On the other hand, the negative correlation between cherry circularity and maximum cherry diameter (−0.60) suggests that highly circular cherries tend to have smaller diameters, an important consideration to avoid undesirable outcomes during selection. According to Silva et al. [18], circularity is sensitive to the elongation of the object and less dependent on the smoothness of its contour; therefore, achieving a balance between circularity and diameter is ideal for obtaining cherries with higher commercial value.

The results of this study corroborate those of Silva et al. [18] in guava, who also reported positive correlations between maximum diameter, minimum diameter, and seed perimeter. These findings suggest that such relationships can be extrapolated to other crops and further emphasize the potential of digital phenotyping for evaluating agronomic traits.

The correlations observed between bean and cherry geometric variables, as extracted from digital analysis equipment, will contribute to the selection of genotypes with larger beans. This is particularly relevant since traits like area and diameter are more difficult to measure accurately by traditional means. According to Marcos Filho [19], simple classification methods, such as measuring bean size using sieves, do not indicate the productive potential of cultivars, which is an important criterion for the commercialization of coffee. In contrast, the digital equipment enables accurate and reliable evaluation of these variables.

The use of the GroundEye^®^ system made it possible to collect high-quality, consistent data free from researcher interference. Moreover, the database generated by the system enables ongoing storage and reanalysis of the information, optimizing the efficiency of breeding programs.

It is therefore recommended that selection efforts focus on increasing area, maximum diameter, and minimum diameter, as these traits not only directly influence productivity but also enhance the commercial appeal of *C. canephora* beans, which are traditionally smaller and less visually attractive in the consumer market. Rectangularity, sphericity, and maximum diameter reflect key aspects of the physiological and morphological development of cherries and beans. Rectangularity is associated with growth and cell division patterns, influencing fruit shape and potentially reflecting genetic and environmental adaptations. Sphericity indicates how closely the fruit or bean approximates an ideal sphere, affecting physical resistance, natural dispersal, and uniformity of ripening. Maximum diameter serves as a direct measure of physical size, related to the storage of essential nutritional reserves that support seed vigor and overall product quality. These traits are thus fundamental not only for plant development, vigor, and adaptation, but also from a commercial standpoint, as they determine bean and cherry size. Larger values in these characteristics correspond to bigger beans and cherries, enabling the use of larger sieves, increasing market value, and enhancing visual appeal, particularly in markets where beans are sold whole.

## 4. Materials and Methods

### 4.1. Experimental Area

The experiment was conducted on a plantation composed of 43 *C. canephora* genotypes, most of which were selected by coffee growers in the region. Planting took place in April 2014 in the municipality of Nova Venécia, in the northern region of Espírito Santo, Brazil, on private property located at latitude 18°66′23″ south and longitude 40°43′07″ west, at an altitude of 199 m above sea level, with an average annual temperature of 23 °C. The region has a tropical climate, characterized by hot and humid summers and dry winters, classified as Aw according to the Köppen classification [20]. The soil is classified as dystric Red-Yellow Latosol (Oxisol), with a clayey texture and undulating relief [16].

The genotypes were arranged in a randomized block design with three replicates, and each treatment was represented by seven plants per genotype. Planting was carried out with a spacing of 3 m between rows and 1 m between plants (3 × 1 m), totaling 3333 plants per hectare. Forty-two genotypes were propagated clonally via cuttings, while only genotype 39 originated from seeds. All plants were cultivated with four stems per plant (Table 6).

Cultivation practices followed the technical guidelines for the crop and consisted basically of weed control with herbicides and mowing, preventive phytosanitary management, liming, fertilization, and drip irrigation [16].

The experimental plots were harvested according to the maturity of each genotype, with an average of four harvest years (2014, 2015, 2016 and 2017). Productivity was measured in liters of ripe fruits and converted to 60 kg bags per hectare, taking into account plant spacing (Partelli et al., 2021 [16]).

### 4.2. Bean Characterization

After harvesting, the cherries were manually pulped, and the beans, still covered with parchment, were placed on plastic trays to dry until they reached 14–15% moisture, determined using a DOLE 500 moisture tester. The parchment was then removed from each bean using a scalpel, after which they were placed in properly labeled paper bags and dried in a forced-air oven until they reached 12% moisture.

### 4.3. Digital Phenotyping of Beans

Using the GroundEye^®^ equipment, four replicates of 50 beans were evaluated for each treatment. The beans were placed on an acrylic tray for image capture. The image analysis system then generated spreadsheets with the results of the following geometric traits: area (B. AREA, cm^2^), circularity (B. CIRCULARITY), maximum diameter (B. MAX D., cm), minimum diameter (B. MIN D., cm), sphericity (B. SPHERICITY), rectangularity (B. RECTANGULARITY), irregularity (B. IRREGULARITY), number of corners (B. N. CORNERS), and perimeter (B. PERIMETER, cm) [22].

### 4.4. Digital Phenotyping of Cherries

Similarly, with the aid of the GroundEye^®^ equipment, four replicates of 50 cherries were evaluated for each genotype. The cherries were arranged on an acrylic tray for image capture. The image analysis system then generated spreadsheets with the results of the following geometric traits: area (C. AREA, cm^2^), circularity (F. CIRCULARITY), maximum diameter (C. MAX DIAMETER), minimum diameter (C. MIN DIAMETER), sphericity (C. SPHERICITY), rectangularity (C. RECTANGULARITY), irregularity (C. IRREGULARITY), and perimeter (C. PERIMETER, cm).

### 4.5. Characterization of the GroundEye^®^ Variables

Area—Represents the amount of space occupied by the surface of an object.

Circularity—A shape factor that is more sensitive to object elongation and less influenced by contour smoothness. A value of 1 corresponds to circular objects, while values less than 1 indicate other shapes, as any other shape with the same maximum diameter has a smaller area.

Maximum Diameter—The longest straight line that passes through the centroid of the sample (bean).

Minimum Diameter—The shortest straight line that passes through the centroid of the sample (bean).

Sphericity of the Shape—Indicates how round an object is. The closer its value is to 12.56, the more closely it resembles a circular shape.

Rectangularity—Represents how rectangular the object’s shape is.

Irregularity of the Contour—Measures the degree of sharpness of the object, based on thermodynamic formalism.

Number of Corners—Determined using the concept of points of interest or points that represent the local maxima in the image function in consideration.

Perimeter—The measurement of the contour of a two-dimensional object, i.e., the sum of all sides of a geometric figure.

### 4.6. Analysis of Variance and Estimates of Genetic Parameters

To assess genetic variability among genotypes, the data obtained through digital phenotyping of beans and cherries were subjected to analysis of variance and estimation of genetic parameters.

The statistical model adopted was:Yij = μ + Gi + Bj + Eijk,
where

Yij = observation related to family i in replicate j; μ = general constant; Gi = effect of family i, (i = 1, 2, …, g); Bj = effect of block j, (j = 1, 2, …, r); and Eijk = experimental error.

From this model, the analysis of variance table was developed, and based on this analysis, estimates of genetic parameters were obtained, namely environmental variance (σ^2^e), phenotypic variance (σ^2^p), genotypic variance (σ^2^g), heritability (h^2^) and the genetic variation index (CVg/CVe), following the expressions proposed by Cruz et al. [23]:

Genotypic variance among genotype means:σ̂g2 = MSG − MSE/r,
where MSG = mean square of genotypes; MSE = mean square error; and r = number of replicates.

Phenotypic variance among genotype means:σ̂p2 = MSG/r

Environmental variance among genotype means:σ̂e2 = MSE/r

Heritability at the genotype mean level:h2 = σ̂g2/σ̂p2

Genetic Variation Index:VI = CVg/CVe,
where

CVg = 100 × Square root of genetic variance/Mean of the traitCVe = 100 × Square root of environmental variance/Mean of the trait.

### 4.7. Ward-MLM

Genetic divergence was quantified using the Ward-MLM (Modified Location Model) method proposed by Franco et al. [14]. First, the Gower index [24] was applied to generate an estimate of the dissimilarity index, which ranges from 0 to 1.

The ideal number of groups was determined based on the pseudo-F and pseudo-T^2^ criteria, combined with the likelihood profile associated with the likelihood ratio test. A logarithmic plot of the likelihood function was then obtained and maximized using the MLM method for different probable numbers of groups. The optimal number of groups for analysis was selected based on the likelihood peaks observed in the plot. Subsequently, a complete MLM analysis was conducted for the number of defined groups, presenting the classification results, including a table describing the groups formed. Canonical analysis was performed for the quantitative variables using the canfile file, which contains the canonical coordinates for the observations. The differences between the groups and the canonical variables (CVs) were analyzed graphically. All analyses were conducted using SAS statistical software version 9.4 M7, and the diagrams were created using SigmaPlot software, version 14.0.

### 4.8. Pearson Correlation

Pearson’s linear correlation coefficient was estimated for the ten variables analyzed, five related to beans and five to cherries. The analysis was performed using the R program version R 4.0.5 and the corrplot package.

The correlation coefficient (r) ranges from −1 to +1: If r = +1, the correlation is perfectly positive; if r = −1, the correlation is perfectly negative; if r = 0, there is no correlation, or the relationship is non-linear.

The magnitude of the correlation coefficient was classified according to Carvalho et al. [24] as follows: Null: r = 0; Weak: 0 < |r| < 0.30; Moderate: 0.30 < |r| < 0.60; Strong: 0.60 < |r| < 0.90; Very strong: 0.90 < |r| < 1; and Perfect: |r| = 1.

## 5. Conclusions

Digital seed analysis successfully detected genetic divergence among the 43 *C. canephora* clones, highlighting its potential as a reliable tool for assessing genetic variability.

The high heritability values observed for all variables indicate that selection is advantageous for all traits analyzed.

Bean sphericity, irregularity, and sharpness were the variables that contributed most to genetic divergence.

Based on the Ward-MLM analysis, crosses between the genotypes of Group I (AD1 and L80) and Group IV (Clementino) are recommended, as they exhibited beans with greater rectangularity and sphericity.

Area, maximum diameter, and minimum diameter are the most suitable variables for selecting genotypes with larger beans.

This geometric analysis approach can be applied to other coffee species, such as *Coffea arabica*, where traits like sieve size directly influence the commercial value of the bean. For example, in the Arabica coffee market, price is often determined by sieve size, with larger beans being more valuable. Therefore, using digital methods to evaluate geometric traits can not only optimize breeding programs but also provide accurate predictions about the commercial value of coffee.

## Figures and Tables

**Figure 1 plants-14-02814-f001:**
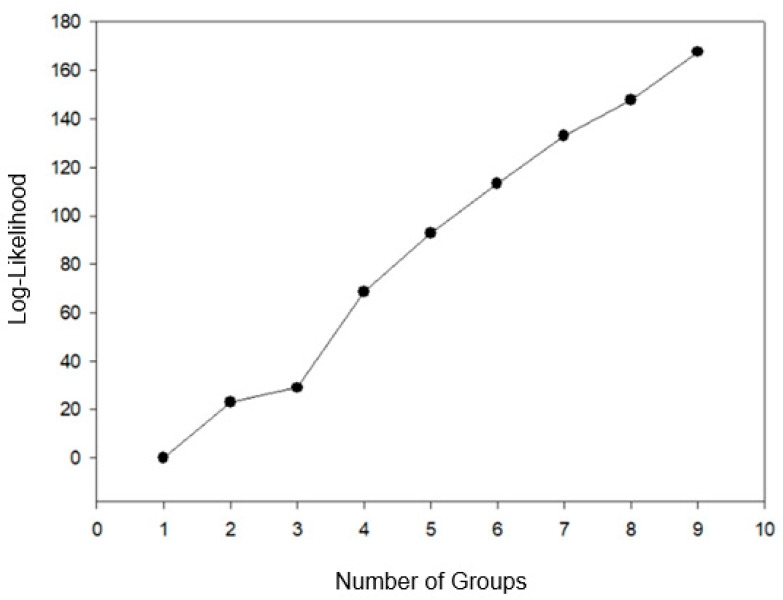
Logarithmic likelihood function (Log-likelihood) as a function of the number of groups formed using the Ward-MLM strategy for *Coffea canephora* beans and cherries.

**Figure 2 plants-14-02814-f002:**
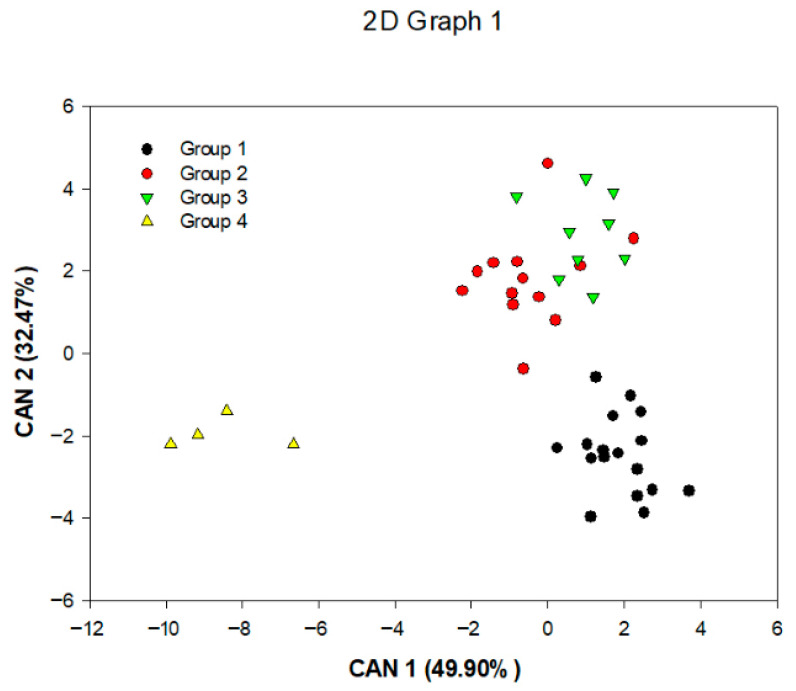
Scatter plot of the first two canonical variables for the four groups formed using the Ward-MLM analysis for *Coffea canephora* beans and cherries.

**Figure 3 plants-14-02814-f003:**
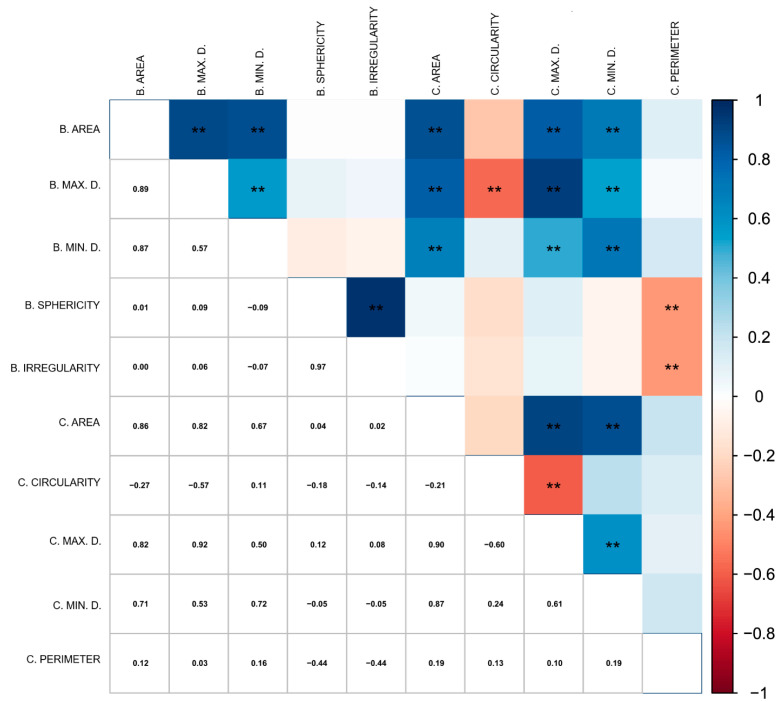
Correlations among ten geometric traits assessed through digital phenotyping (B. AREA: bean area; B. MAX. D.: bean maximum diameter; B. MIN. D.: bean minimum diameter; B. SPHERICITY: bean sphericity; B. IRREGULARITY: bean irregularity; C. AREA: cherry area; C. CIRCULARITY: cherry circularity; C. MAX. D.: cherry maximum diameter; C. MIN. D.: cherry minimum diameter; C. PERIMETER: cherry perimeter) of 43 *C. canephora* genotypes. (** correspond to significance at *p* < 0.01).

**Table 1 plants-14-02814-t001:** Summary of analysis of variance for geometric variations in *Coffea canephora* beans.

SV	DF	Mean Square								
		AREA	CIRC.	MAX. D.	MIN. D.	SPHER.	RECT.	IRREG.	N. CORN.	PERI.
BLOCKS	3	0.203	0.0032	0.0002	0.00019	46.926	0.00007	0.00092	899.858	0.088
GENOTYPES	42	0.334 **	0.0655 **	0.0834 **	0.0426 **	126.24 **	0.0575 **	0.0039 **	14,566.632 **	1.148 **
RESIDUAL	126	0.114	0.0008	0.00016	0.00013	17.841	0.00001	0.00053	439.630	0.041
TOTAL	171									
CV%		8.048	1.25	1.612	1.994	20.644	0.5190	30.043	11.101	7.543

SV—source of variation; DF—degrees of freedom; GEN.—genotype; RES.—residual; AREA—area; CIRC.—circularity; MAX. D.—maximum diameter; MIN. D.—minimum diameter; SPHER.—sphericity; RECT.—rectangularity; IRREG.—irregularity; N. CORN.—number of corners; PERI.—perimeter; CV—coefficient of variation. (** correspond to significance at *p* < 0.01).

**Table 2 plants-14-02814-t002:** Summary of analysis of variance for geometric variations in *Coffea canephora* cherry.

SV	DF	Mean Square							
		AREA	CIRC.	MAX. D.	MIN. D.	SPHER.	RECT.	IRREG.	PERI.
BLOCKS	3	0.0203	0.00242	0.0024	0.0040	391.160	0.00016	0.0049	2.7311
GENOTYPES	42	0.3348 **	0.0810 **	0.2619 **	0.1624 **	4691.685 **	0.0558 **	0.0028 **	29.3073 **
RESIDUAL	126	0.0114	0.0016	0.0040	0.0034	1357.386	0.00029	0.0035	5.2493
TOTAL	171								
CV%		8.048	4.914	4.554	5.055	62.755	2.292	28.710	27.950

SV—source of variation; DF—degrees of freedom; AREA—area; CIRC.—circularity; MAX. D.—maximum diameter; MIN. D.—minimum diameter; SPHER.—sphericity; RECT.—rectangularity; IRREG.—irregularity; PERI.—perimeter; CV—coefficient of variation. (** correspond to significance at *p* < 0.01).

**Table 3 plants-14-02814-t003:** Means of quantitative variables for the three groups formed by the Ward-MLM method and the two canonical variables, in *Coffea canephora* beans and cherries.

Variable	Group				Can	
	I	II	III	IV	CAN I	CAN II
B. AREA	0.3512	0.4292	0.4211	0.3175	0.173131	0.69723
B. CIRCULARITY	0.7506	0.7731	0.68	0.7925	−0.311236	−0.297117
B. MAX. D.	0.7718	0.8415	0.89	0.7125	0.29108	0.746936
B. MIN. D	0.5676	0.6346	0.5933	0.555	0.037303	0.498843
B. SPHERICITY	17.8841	20.9515	24.0800	27.7150	−0.562638	0.369202
B. RECTANGULARITY	0.7906	0.7869	0.7833	0.7825	0.371059	−0.418348
B. IRREGULARITY	0.0576	0.0838	0.0978	0.12	−0.576385	0.439309
B. N. EDGES	195.669	191.7046	192.6933	167.1575	0.155414	0.010412
B. PERIMETER	24.688	29.185	30.844	28.200	−0.250193	0.7832
B. SHARPNESS	0.7529	0.68	0.6256	0.5875	0.501549	−0.472493
C. AREA	12.718	14.862	14.478	11.300	0.2077	0.573342
C. CIRCULARITY	0.86	0.8531	0.7644	0.885	−0.218075	−0.489773
C. MAX. D.	13.741	14.908	15.556	12.775	0.262336	0.67339
C. MIN. D.	11.606	12.523	11.722	11.100	0.116307	0.34097
C. SPHERICITY	75.9541	70.9931	37.1989	22.2325	0.38711	−0.224035
C. RECTANGULARITY	0.7724	0.7685	0.7767	0.7825	−0.20564	−0.054719
C. IRREGULARITY	0.2453	0.2562	0.1556	0.1075	0.419295	−0.135833
C. PERIMETER	90.976	98.108	68.522	49.400	0.418204	−0.039292

**Table 4 plants-14-02814-t004:** Identification of the genotypes of the four groups (GI, GII, GIII, and GIV), formed by the Ward-MLM strategy and average yield in liters of coffee per 60 kg bag and productivity of *Coffea canephora* genotypes.

Group	Genotype	Yield	Productivity
		Liters/Bag	Bags/ha
I	Verdim R	398.32	82.35
	B01	418.21	43.11
	AD1	303.49	128.93
	L80	364.68	105.53
	Bamburral	357.23	86.6
	Z35	402.23	72.48
	Z29	366.61	70.15
	Z37	363.78	86.84
	Z36	316.62	91.45
	18	422.67	49.3
	Tardio C	345.15	74.73
	P2	316.7	92.32
	122	330.03	81.07
	Verdim D	336.65	90.43
	Sementes	218.17	70.32
	Ouro Negro 1	359.87	72.23
	Ouro Negro 2	316.46	74.7
II	Beira rio 8	439.72	61.81
	Tardio V	329.95	69.84
	AP	313.53	86.28
	Z39	339.92	95.48
	Z40	356.18	74.55
	Z18	330.2	80.05
	Z21	294.01	102.49
	Ouro Negro	346.39	75.24
	Cheique	372.10	70.77
	Emcapa 02	323.18	97.1
	P1	332.5	93.47
	LB1	312.21	118.08
	Emcapa 143	312.26	94.65
III	Bicudo	398.32	82.35
	700	303.23	89.19
	CH1	359.87	81.40
	Imbigudinho	324.82	79.67
	Graudão HP	317.59	86.13
	Pirata	379.63	76.22
	Peneirão	326.6	99.22
	A1	365.44	108.19
	Emcapa 153	322.45	85.52
IV	Alecrim	341.46	54.37
	Valcir P	341.11	87.93
	Z38	372.53	74.94
	Clementino	355.48	97.86

**Table 5 plants-14-02814-t005:** Distance between the groups formed by the Ward-MLM procedure for *Coffea canephora* beans and cherries, proposed by Franco et al. (1998) [14].

Group	I	II	III	IV
I	0			
II	2,942,499	0		
III	3,538,909	2,587,098	0	
IV	10,988,678	9,027,114	11,471,734	0

**Table 6 plants-14-02814-t006:** Identification of the 43 genotypes of *Coffea canephora*. Nova Venécia—ES, Brazil, 2022.

ID	NAME	ID	NAME	ID	NAME
**1**	Verdim R	**15**	Bamburral	**29**	Tardio C
**2**	B01	**16**	Pirata	**30**	A1
**3**	Bicudo	**17**	Peneirão	**31**	Cheique
**4**	Alecrim	**18**	Z39	**32**	P2
**5**	700	**19**	Z35	**33**	Emcapa 02
**6**	CH1	**20**	Z40	**34**	Emcapa 153
**7**	Imbigudinho	**21**	Z29	**35**	P1
**8**	AD1	**22**	Z38	**36**	LB1
**9**	Graudão HP	**23**	Z18	**37**	122
**10**	Valcir P	**24**	Z37	**38**	Verdim D
**11**	Beira Rio 8	**25**	Z21	**39**	Sementes
**12**	Tardio V	**26**	Z36	**40**	Emcapa 143
**13**	AP	**27**	Ouro Negro	**41**	Ouro negro 1
**14**	L80	**28**	18	**42**	Ouro negro 2
				**43**	Clementino

Genotype 33 belongs to cultivar Emcapa 8111 and genotypes 34 and 39 to cultivar Emcapa 8131 [21]. Genotypes 30 and 35 belong to cultivar Andina [17]; 8, 7, 13, 17, 32, and 36 to cultivar Monte Pascoal [18]; and 1, 11, 15, 16, 30, and 43 to cultivar Tributun [18,19].

## Data Availability

The study does not report any data.

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
