# Peer review of "Genetic Diversity in Coffea canephora Genotypes via Digital Phenotyping"

_plants, 2025, doi:10.3390/plants14182814_

Round 1

Reviewer 1 Report

Comments and Suggestions for Authors
  1. Materials and Methods, lines 296-297. The authors indicated that 42 genotypes were propagated clonally and only 1 by seed. Do they refer to the origin of the genetic material or was such propagation carried out during the experiment?
  2. Materials and Methods. The information provided in lines 303-305 on the name of cultivars used does not fully correspond to the information provided in Table 6. Please make such information congruent.
  3. Materials and Methods. Please report the units (or otherwise the formulas used) to determine composite variables such as CIRCULARITY, SPHERICITY, RECTANGULARITY and IRREGULARITY in Digital phenotyping of both, beans (lines 316-318) and cherries (lines 324-327).
  4. Materials and Methods. Please apply uniform terminology. The groups of individuals of each cultivar were sometimes called genotype (e.g. lines 284-285, 302) and in other they are called as families (e.g. line 354).
  5. Materials and Methods. The layout of the experiment is too simple to obtain reliable estimates of genetic parameters. For example, it is not possible to obtain a reliable environmental variance when only one environment was tested. In order to obtain reliable estimates of variance, heritability and genetic variation indices, mating designs should be used. It is suggested to remove the genetic parameters reported at the bottom of Tables 1 and 2 and related text (including second conclusion). The only information worth considering from the analysis of variance is the determination that there are statistical differences between genotypes and the magnitude of mean squares as indicators of phenotypic variance.
  6. Traits Yield and Productivity are reported in Table 4; however, such characteristics are not mentioned in the Materials and Methods section.

Author Response

• Materials and Methods, lines 296-297: The authors indicated that 42 genotypes were propagated clonally and only 1 by seed. Are they referring to the origin of the genetic material, or was this propagation performed during the experiment?
The term refers to the origin of the genetic material.
• Materials and Methods: The information provided in lines 303-305 regarding the names of the cultivars used does not fully correspond to the information presented in Table 6. Please make this information consistent.

The information has been corrected, including in Table 4.
• Materials and Methods: Please provide the units (or, alternatively, the formulas used) to determine composite variables such as CIRCULARITY, SPHERE, RECTANGULARITY, and IRREGULARITY in digital phenotyping, both for grains (lines 316-318) and fruits (lines 324-327).

Units were corrected in the text as requested by the reviewer.
• Materials and Methods: Please apply uniform terminology. Groups of individuals of each cultivar are sometimes called genotypes (e.g., lines 284-285, 302) and sometimes called families (e.g., line 354).
The correction was made as recommended by the reviewers and standardized to 'genotype'.
• Materials and Methods: The experimental design is too simple to obtain reliable estimates of genetic parameters. For example, it is not possible to obtain reliable environmental variance when only one environment was tested. To obtain reliable estimates of variance, heritability, and indices of genetic variation, cross designs should be used. It is suggested that the genetic parameters presented at the end of Tables 1 and 2 and the related text (including the second conclusion) be removed. The only relevant information from the analysis of variance is the finding that there are statistical differences between the genotypes and the magnitude of the mean squares as indicators of phenotypic variance.
The text was removed as recommended by the reviewer.
• The Yield and Productivity characteristics are presented in Table 4; however, these characteristics are not mentioned in the Materials and Methods section.
Because these data came from previous studies. Partelli (2021).

Reviewer 2 Report

Comments and Suggestions for Authors

The manuscript presents a clear and relevant study using digital phenotyping to evaluate genetic diversity in Coffea canephora genotypes. It effectively employs quantitative trait analysis, correlation studies, and multivariate clustering (Ward-MLM) to propose promising genotypes for breeding programs focused on bean size and quality. The writing is mostly clear, though minor language and structural polishing will significantly improve clarity and readability. The study is technically sound, but there is room for enhancing the discussion depth, particularly in linking phenotyping data with breeding outcomes.

  1. The current title is accurate but lacks appeal. Consider a slightly refined title.
  2. Consider adding more relevant keywords for instance:
    Genetic divergence, bean geometry, high-throughput phenotyping, coffee breeding, heritability, Ward-MLM
  3. First sentence in the abstract reads like a fact rather than linking to the research question.

  4. Use of “favorably” and “effectively” should be supported with precise data.

  5. In the introduction section please provide a stronger rationale for digital phenotyping.
  6. The discussion is largely descriptive—more interpretation of biological relevance is needed. 
  7. Link specific bean traits (e.g., rectangularity, sphericity) with market value or processing outcomes.

  8. No mention of limitations or future perspectives.

Results:

9. Consider restructuring the section into the following subsections with bold or numbered titles for clarity:

2.1 Genetic Parameter Estimation
2.2 Genetic Divergence and Grouping via Ward-MLM
2.3 Phenotypic Correlations between Bean and Cherry Traits
2.4 Implications for Genotype Selection

10. Much of the text is descriptive, focusing on statistical outputs (variance, clustering, correlation) without connecting them to biological or breeding implications.
For instance, explain why traits like rectangularity, sphericity, and maximum diameter are important from the physiological or market value perspective.

11. The authors discuss the genotype grouping (I to IV) in a good manner but, the implications for breeding programs are under-discussed. I suggest to explicitly relate trait patterns in each group to genetic gain potential. For instance, comment that Group IV genotypes could be ideal donors for increasing sphericity, while Group I lines offer higher productivity potential.

Sentences like the following can be added. 
“Crosses between Group I (high-yielding, angular beans) and Group IV (high circularity and sphericity) genotypes may generate transgressive segregants combining favorable productivity and bean geometry traits.”

conclusion:

Consider adding one sentence about how this approach could be applied to other coffee species or extended to predict quality traits.

language; 

Use active voice more frequently.
Avoid repetitive use of terms like “genotypes” or “genetic variability”vary the phrasing.

Author Response

The manuscript presents a clear and relevant study that uses digital phenotyping to assess genetic diversity in Coffea canephora genotypes. The work effectively employs quantitative trait analysis, correlation studies, and multivariate clustering (Ward-MLM) to propose promising genotypes for breeding programs focused on bean size and quality. The writing is, for the most part, clear, although minor adjustments to language and structure could significantly improve clarity and readability. The study is technically sound, but there is room for further discussion, especially when relating phenotyping data to breeding results.
1. The current title is adequate but lacks appeal. Consider a slightly more refined title.
• We appreciate the suggestion regarding the manuscript title. However, after careful consideration, we prefer to maintain the current title, as we believe it adequately reflects the content and focus of the study.

2. Consider adding more relevant keywords, for example:
Genetic divergence, bean geometry, high-yield phenotyping, coffee breeding, heritability, Ward-MLM.
• Keywords were added as suggested by the reviewer.
3. The first sentence of the abstract reads as a fact, rather than directly connecting to the research question.
• The first sentence of the abstract was removed as recommended by the reviewer.
4. The use of the terms "favorably" and "effectively" should be supported by accurate data.
Thank you for your observation. The terms "favorably" and "effectively" have been replaced, maintaining the original meaning of the text and avoiding generalizations, as suggested.

In the introduction, please provide a more robust justification for the use of digital phenotyping.
• This paragraph has been added to the text.
• "Digital phenotyping offers significant advantages over traditional methods by enabling objective, standardized, and large-scale analysis of morphological traits that would be difficult, time-consuming, or prone to errors when assessed manually. This technology allows for the collection of precise and reproducible data, accelerating the screening and selection of genotypes with superior potential in breeding programs. Furthermore, digital phenotyping reduces evaluation costs and time, contributing to the integration of phenotypic information with genomic data in modern plant breeding strategies."
5. The discussion is mostly descriptive—more interpretation of biological relevance is needed.
We revised the discussion section to include a more interpretive analysis, emphasizing the biological relevance of the results, their implications for genetic improvement, and the potential use of the variability observed among the genotypes studied.
6. Relate specific grain characteristics (e.g., rectangularity, sphericity) to market value or processing results.

Paragraph inserted at the end of the discussion:
These characteristics are directly related to the larger size of the grains and pods, which implies a larger sieve and, consequently, a higher market value. Furthermore, larger grains are visually more attractive, especially in markets where they are sold whole.

7. No mention of limitations or future prospects.
We acknowledge that the manuscript does not currently include a section on study limitations or future prospects. We will consider including this information in future versions of the work to strengthen the interpretation of the results and guide subsequent research.

Results:
9. Consider restructuring the section into subsections with bold or numbered titles for clarity:
2.1 Estimation of Genetic Parameters
2.2 Genetic Divergence and Clustering via Ward-MLM
2.3 Phenotypic Correlations between Grain and Fruit Traits
2.4 Implications for Genotype Selection
• Restructured section
10. Much of the text is descriptive, focusing on statistical results (variance, clustering, correlation) without connecting them to biological or breeding implications.
For example, explain why traits such as rectangularity, sphericity, and maximum diameter are important from a physiological or market value perspective.
We appreciate the comment and agree that the biological implications of morphological traits deserve greater emphasis. The information below has been added to the end of the discussion.
Rectangularity, sphericity, and maximum diameter are characteristics that reflect important aspects of the physiological and morphological development of the fruit and grain. Rectangularity is associated with the pattern of growth and cell division, influencing the shape of the fruit, which may be related to genetic and environmental adaptations. Sphericity indicates how close the shape of the fruit or grain is to an ideal sphere, a factor that can affect physical strength.

The natural dispersion and uniformity of fruit ripening. Maximum diameter, in turn, is a direct indicator of physical size, which is related to the storage capacity of nutritional reserves essential for seed vigor and the quality of the final product.
Thus, these characteristics not only influence the commercial value of coffee but also reflect biological processes fundamental to the development, vigor, and adaptation of plants.
Traits such as rectangularity, sphericity, and maximum diameter are important from a market value perspective because they are used to measure bean and fruit size. Thus, the higher the values of these characteristics, the larger the bean and fruit size, which is directly associated with a higher commercial value of the coffee.

11. The authors discuss the grouping of genotypes (I to IV) adequately, but the implications for breeding programs are little explored. I suggest explicitly relating the trait patterns in each group to the potential for genetic gain. For example, comment that Group IV genotypes could be ideal donors for increased sphericity, while Group I lines offer greater productivity potential.
Sentences such as:
"Crosses between Group I genotypes (high-yielding angular beans) and Group IV genotypes (high circularity and sphericity) can generate transgressive segregants that combine favorable productivity and bean geometry traits."
Sentence added to discussion
Conclusion:
Consider adding a sentence about how this approach could be applied to other coffee species or extended to predict quality traits.
Paragraph added to conclusion:
This geometric analysis approach can be applied to other coffee species, such as Coffea arabica, where traits like sieve size directly influence the commercial value of the bean. For example, in the Arabica coffee market, price is often determined by sieve size, with larger beans being more valuable. Therefore, using digital methods to evaluate geometric traits can not only optimize breeding programs but also provide accurate predictions about the commercial value of coffee.
Language:
Use active voice more frequently.
Avoid repetitive use of terms like "genotypes" or "genetic variability"; vary the expression.
Thank you for the suggestion. We chose to maintain the current voice and the consistent use of terms like "genotypes" and "genetic variability" to preserve the clarity and scientific accuracy of the manuscript.

Reviewer 3 Report

Comments and Suggestions for Authors

The manuscript entitled " Genetic diversity in Coffea canephora genotypes via digital phenotyping" presents a study on the variability of 43 canephora clones which based on several morphological traits of cherries and beans. These features were measured automatically using an optical technology, and the features were combined into indices and compared for group dissimilarity using Ward-MLM analysis. Additionally, the authors provide a correlation analysis of measured traits. Based on the estimation of genetic and environmental variance, traits affected by the genetic component of variability to the greatest extent are indicated.

The manuscript is well-prepared and easy to read, with well-defined aims, and the idea and methodology leave no doubt about their correctness.

It is suggested that using this convenient, highly objective, and time-saving method may have a perspective for application in breeding. However, relevant features impacting product quality are not necessarily determined by the morphological properties of fruits. Thus, the selection of desirable genotypes and breeding oriented only on morphological traits does not necessarily have to be effective in plant improvement.

Although I like this paper due to its simplicity and clear presentation of results, I do not find a significant impact of the research on understanding plant biology, particularly in genetic aspects. The occurrence of variability of traits for each plant organ and agricultural product compound is obvious, particularly in polygenically inherited traits as size and shape. It could be interesting as an introductory part of a larger paper, which is followed by more specific non-morphological analysis. It may be a valuable proposition for breeders, and thus, I suggest submitting the manuscript to a journal more oriented toward plant breeding or practical agronomy. The manuscript could also be submitted as a short communication, after moving a large part of the Tables content to a supplement. The manuscript would gain in quality if the text and presentation of results (large tables) were more concise.

Author Response

The manuscript, titled "Genetic Diversity in Coffea canephora Genotypes via Digital Phenotyping," presents a study of the variability of 43 canephora clones, based on several morphological characteristics of fruits (cherries) and beans. These characteristics were measured automatically using optical technology, and the data were combined into indices and compared to assess dissimilarity between groups using Ward-MLM analysis. Additionally, the authors present a correlation analysis of the measured traits. Based on the estimated genetic and environmental variances, they indicate the traits most influenced by the genetic component of variability.
The manuscript is well-designed and easy to read, with well-defined objectives, and the idea and methodology leave no doubt as to their correctness.
It is suggested that the use of this convenient, highly objective, and time-saving method may have potential for application in breeding programs. However, relevant traits that impact product quality are not necessarily determined by the morphological properties of the fruits. Therefore, selecting desirable genotypes and breeding based solely on morphological characteristics will not necessarily be effective in improving plants.
Thank you for your observation. We would like to clarify that, in the case of coffee, the parts with the greatest commercial value are precisely the bean and the fruit. Therefore, evaluating the morphological characteristics of these structures, such as size, rectangularity, and sphericity, is directly related to the quality and value of the product, making it relevant and effective for breeding programs.
While I appreciate this article for its simplicity and clear presentation of results, I do not see a significant impact of this research on the understanding of plant biology, particularly genetic aspects. The occurrence of trait variability for each plant organ and agricultural product component is obvious, especially in polygenic traits such as size and shape. It could be interesting as an introductory part of a larger work, followed by more specific, non-morphological analyses. It could be a valuable proposal for breeders, and therefore, I suggest submitting the manuscript to a journal more focused on plant breeding or practical agronomy. The manuscript could also be submitted as a short paper, after transferring much of the table content to a supplement. The manuscript would improve in quality if the text and presentation of results (extended tables) were more concise.

We appreciate the constructive comments and suggestions provided. We recognize the importance of the observations regarding the study's impact on the understanding of plant biology and the need to expand the discussion with more specific analyses. We will also consider recommendations regarding the conciseness of the text and presentation of results, especially regarding the extended tables.

Based on the contributions of all reviewers, we revised the manuscript to improve the clarity, focus, and relevance of the work, while maintaining alignment with the journal's scope.

Round 2

Reviewer 1 Report

Comments and Suggestions for Authors
  1. Please delete the abbreviations at the footer of Tables 1 and 2 corresponding to deleted information (genetic parameters).

Author Response

Abbreviations have been removed.

Reviewer 2 Report

Comments and Suggestions for Authors

The authors did a good job with addressing the previous issues in the revision. I feel the mansucript is now fit for publication in its current form.

Author Response

Thanks.

Reviewer 3 Report

Comments and Suggestions for Authors

The revised version shows improvement in the form of presentation and discussion of the results, which allows me to recommend the manuscript for publication in "Plants".

Author Response

Thanks